# The Bidirectional Link between Nutritional Factors and Inflammatory Bowel Diseases: Dietary Deficits, Habits, and Recommended Interventions—A Narrative Review

**DOI:** 10.3390/foods12101987

**Published:** 2023-05-14

**Authors:** Ilaria Maria Saracino, Enzo Spisni, Veronica Imbesi, Chiara Ricci, Nikolas Konstantine Dussias, Patrizia Alvisi, Paolo Gionchetti, Fernando Rizzello, Maria Chiara Valerii

**Affiliations:** 1Microbiology Unit, IRCCS, Azienda Ospedaliero-Universitaria di Bologna, University of Bologna, Via Massarenti 9, 40138 Bologna, Italy; ilariamaria.saracino@studio.unibo.it; 2Department of Biological, Geological and Environmental Sciences, University of Bologna, Via Selmi 3, 40126 Bologna, Italy; mariachiara.valerii2@unibo.it; 3Department of Medical and Surgical and Sciences, University of Bologna, Via Massarenti 9, 40138 Bologna, Italy; 4Gastroenterology Unit, ASST Spedali Civili di Brescia, University of Brescia, Piazza del Mercato 15, 25121 Brescia, Italy; chiara.ricci@unibs.it; 5IBD Unit, IRCCS, Azienda Ospedaliero-Universitaria di Bologna, University of Bologna, Via Massarenti 9, 40138 Bologna, Italy; nikolas.dussias@studio.unibo.it (N.K.D.); paolo.gionchetti@unibo.it (P.G.); fernando.rizzello@unibo.it (F.R.); 6Pediatric Unit, Maggiore Hospital, Largo Bartolo Nigrisoli, 2, 40133 Bologna, Italy; patrizia.alvisi@ausl.bologna.it

**Keywords:** inflammatory bowel diseases, Crohn’s disease, ulcerative colitis, diet, malnutrition, nutritional interventions, enteral nutrition

## Abstract

Inflammatory bowel diseases comprise Crohn’s disease and ulcerative colitis, two chronic inflammatory disorders of the digestive tract that develop in adolescence and early adulthood and show a rising pattern in industrialized societies, as well as in developing countries, being strongly influenced by environmental pressures such as nutrition, pollution and lifestyle behaviors. Here, we provide a narrative review of the bidirectional link between nutritional factors and IBD, of dietary deficits observed in IBD patients due to both the disease itself and dietary habits, and of the suggested nutritional interventions. Research of the literature was conducted. Clinical and basic research studies consistently demonstrate that diet could alter the risk of developing IBD in predisposed individuals. On the other hand, dietary interventions represent a valid tool in support of conventional therapies to control IBD symptoms, rebalance states of malnutrition, promote/maintain clinical remission and improve patients’ quality of life. Although there are no official dietary guidelines for patients with IBD, they should receive nutritional advice and undergo oral, enteral, or parenteral nutritional supplementation if needed. However, the dietary management of malnutrition in IBD patients is complex; future clinical studies are required to standardize its management.

## 1. Introduction

Epidemiologically we are now in the age of noncommunicable degenerative and human-influenced diseases that arise in industrialized and urbanized societies and are strongly influenced by environmental pressures such as nutrition, pollution and lifestyle behaviors. Inflammatory bowel diseases (IBD) are the epitome of these types of pathologies. IBD comprise Crohn’s disease (CD) and ulcerative colitis (UC), two chronic, inflammatory disorders of the digestive tract that develop in adolescence and early adulthood and affect about 1.5 million Americans and 2.2 million European patients [1]. IBD are characterized by an uncontrolled immune-mediated inflammatory response in genetically predisposed individuals exposed to environmental factors that include diet, medications, nicotine, infectious agents, stress, pollution and lifestyle, which together contribute to the trigger of the gut chronic inflammatory loop [2]. The disease burden is high, with prevalence surpassing 0.3% in North America, Oceania, and most countries in Europe. By contrast, countries in Africa, Asia, and South America, whose societies are becoming increasingly westernised and urbanised, are mirroring the progression of IBD in the Western world during the 1900s [3]. In an interesting perspective published in 2021 by Kaplan and collaborators [4], authors described the epidemiological stages of IBD in three main areas of the world: “developing countries”, “newly industrialized countries” and “Western world” (as defined by the United Nations) [5]. In the same paper, authors described the evolution of IBD across three distinct epidemiological stages: emergence, acceleration in incidence and compounding prevalence. Emergence is represented by sporadic cases of IBD; acceleration in incidence indicates a steadily increase in incidence, with a low prevalence [6]; compounding prevalence reflects a steady increase in IBD prevalence, despite a lowering or stabilization in incidence [7]. These stages reflect the nature of IBD: a chronic disease that can occur in individuals of any age, but is most commonly diagnosed in adolescents and young adults (ages 18–35 years) who have a long life expectancy [6,8]. The first burden in the world evolution of IBD occurred in North America and Western Europe following the advent of the industrial revolution [4], and by the 1950s, the Western world was shifting to the second epidemiological stage that spanned the second half of the 20th century [6]. Most countries of the Western world transitioned into the third stage in the evolution of IBD by the end of the 20th century, with a concomitant increase in pediatric IBD [9]. In the same period, newly industrialized countries in Asia and Latin America entered the second stage; if these regions will follow the epidemiological patterns of the Western world, they are likely to transit into the third stage over the next three decades. Developing countries are currently in the first stage of evolution and can expect to enter in the second stage if their societies westernize. Analyzing epidemiological data and making appropriate projections, it has been estimated that over the next decade, the prevalence of IBD in the Western world will be very close to 1% [4]. Several reviews have compared the environmental determinants of IBD in the Western world with those of newly industrialized countries [10,11,12]. For example, over the past 30 years, China experienced a massive population shift from rural communities to megacities, and inhabitants in these urban environments have been exposed to the westernization of their society including diets, pollution and lifestyle behaviors [13]. As a result of that, some regions in China have already reached IBD incidence values that are close to those of the Western world. Moreover, many studies have concluded that people emigrating from low-prevalence regions such as Asia to high-prevalence countries such as northern Europe are at greater risk of developing IBD. This is especially true for first-generation immigrant children who show similar IBD incidence and prevalence patterns to the indigenous population [14].

The rising pattern of IBD has provided a unique opportunity for researchers to focus on identifying the environmental risk factors contributing to its pathogenesis. Among them, until now, the most consistent results have been obtained for the correlation between diet and IBD prevalence, although it cannot be excluded that environmental pollutants will be identified as major determinants in the future.

### 1.1. Diets: The Good and the Bad

Mediterranean, Indian, Japanese and Southeast Asian traditional diets are all considered as healthy diets. In the 1960s, the term “Mediterranean diet” was introduced by the physiologist Ancel Keys, defining the diet followed by populations living along the Mediterranean Basin. These populations showed a lower mortality rate and incidence of cardiovascular disease than other populations. The Mediterranean diet (Figure 1) has been proven to enrich beneficial gut bacteria, support gut barrier function and reduce inflammation, thus acting as a preventive factor for the onset of IBD [15,16,17]. Traditional Indian [18], Japanese [19] and Southeast Asian [20] diets are basically similar to the Mediterranean diet, relying on seafood and plant foods such as vegetables, legumes, fruits, herbs and spices, as well as on a low consumption of meat, sugar and highly processed foods. These diets are naturally rich in minerals (e.g., calcium, potassium, iron, magnesium), vitamins, fiber and long-chain omega-3 fatty acids. Spices not only enhance the flavor and taste of foods, but are also considered as functional foods carrying biologically active phytochemicals such as curcumin, gingerols and essential oils, all known for their anti-inflammatory, antibacterial and anticancer properties [21].

A Western-style dietary habit (Figure 1) is instead considered an important risk factor for IBD [17]. The so-called Western diet is characterized by the consumption of high amounts of animal proteins, saturated fats and processed foods (rich in additives, emulsifiers and food preservatives) and low amounts of vegetables, unsaturated fats, fibers and fruits. The Western diet is also characterized by an unbalanced ratio of polyunsaturated fatty acids (n-3/n-6 PUFAs). Linoleic acid (a n-6 PUFA) consumption has markedly increased during the 20th century [22] and is associated with an increased risk of UC, whereas oleic acid (n-9 monounsaturated fatty acid or MUFA) or n-3 PUFAs consumption is beneficial. The low intake of fruit and vegetables is detrimental too, since it is well known that a long-term high intake of dietary fiber reduces CD risk [23,24,25,26].

### 1.2. FAO Survey on the “Nutrition Transition”

In the survey of food consumption, FAO addressed the “nutrition transition” in the world, mainly in developing countries, stating that “it is contributing to the causal factors underlying noncommunicable diseases even in the poorest countries”. The adverse dietary changes include shifts in the structure of the diet towards a higher energy density food with a greater role for fat and added sugars, greater saturated fat intake (mostly from animal sources), reduced intakes of whole grains and dietary fibers, and reduced fruit and vegetable intakes. Additionally, the increase in urbanization increased the distances of people from primary food production, leading to the use of processed long-life foods. This nutritional shift has taken place in South America (especially Brazil), East Asia (especially China) and the Near East/North Africa region, while some parts of Southeast Asia and Sub-Saharan Africa are still basically tied to traditional low-calorie, low-fat diets [27,28].

All these observations are consistent with the IBD epidemiological map (Figure 2).

Here, we provide a narrative review of the bidirectional link between nutritional factors and IBD, of dietary deficits observed in IBD patients due to both the disease itself and dietary habits, and of the desirable nutritional interventions. Because there are still no validated tools for screening malnutrition in patients with IBD and no standardized nutritional regimen for these patients, it is important to summarize the available studies and their results in order to increasingly refine these aspects of IBD patient management.

## 2. Materials and Methods

In order to provide an overview of research focused on diet, malnutrition and nutritional interventions in IBD patients, literature research was conducted on PubMed Central with “advanced” and “MeSH” tools, and links for articles were accessed with University of Bologna login.

Term queries: (((inflammatory bowel disease) AND (diet*ther* OR nutri*inter*)) OR ((inflammatory disease AND (nutri*deficienc* OR malnutrition))); (((inflammatory bowel disease) AND ((diet) OR (western diet) OR (nutri*))). Inclusion criteria were as follows: clinical trial, observational study (for the food habits-related causes of malnutrition), basic research (only for the latter term query), English language, full text available, published in the last 10 years (2012–2022). Exclusion criteria were as follows: clinical case studies, letters or editorials of magazines, abstracts to conferences, book chapters. The Grading of Recommendations Assessment, Development and Evaluation (GRADE) approach was not used in the study selection process. The final search included 155 articles, from which 83 abstracts were selected. From these abstracts, the 57 most relevant studies were selected for the complete text reading and analyzed in this review. The results are exposed in three main areas: (1) food-related pathogenetic molecular mechanisms—dietary risk factors, dietary protective factors; (2) malnutrition in IBD—macro and micronutrient deficiency prevalence in IBD; disease-related causes of malnutrition in IBD, food-habits-related causes of malnutrition in IBD; (3) nutritional interventions in IBD. PRISMA flow diagram is reported in Figure 3.

## 3. Results

### 3.1. Food-Related Pathogenetic Molecular Mechanisms of IBD

In IBD, altered gut microbial signatures have been consistently reported, and there is little doubt that industrialization goes along with alteration of the intestinal microbial ecology in humans. For example, metagenomic studies in stool specimens of IBD patients clearly indicate a reduction in ecological diversity with a lower proportion of Firmicutes and an increased abundance of Proteobacteria and Bacteroidetes [29]. Dietary habits may result in an altered susceptibility to IBD in a variety of ways; it has been proven that food components can modify intestinal permeability (e.g., short chain fatty acids—SCFAs, vitamin D, vitamin A, zinc, cysteine, methionine, glutamine, tryptophan and arginine decrease intestinal permeability, while gluten, glucose, fructose, fats, ethanol and emulsifiers increase it) and can serve as ligands for various receptors expressed in enterocytes (e.g., farnesoid X receptor, aryl hydrocarbon receptor, pregnane X receptor, and specific G protein-coupled receptors) [30,31]. A family of intracellular danger-sensing molecules called NLRs (NOD-like receptors) has been implicated in the pathogenesis of IBD. Within this family are the NOD (nucleotide-binding oligomerization domain) and NLRP (pyrin domain-containing NLR) subfamilies. The link between NOD2 mutation and CD has been well established, and it is the most common genetic variant associated with CD [32]. In addition, alterations in another family of bacterial sensor genes, the Toll-like receptors (TLRs), have been well described in IBD. It is noteworthy that the Western diet increases both NOD-2 and TLR-5 mRNA levels in experimental CD models [33].

The role of diet in shaping gut microbiota is well established. Thus, interactions between intestinal microbes and food can in turn influence IBD pathogenesis. Defective innate immunity could contribute to IBD pathogenesis by enabling a tolerance breakdown to a normal intestinal microbiota in genetically predisposed individuals. These findings demonstrate the innate immunity control of the adaptive immune response and the crucial role of the gene–environment interplay in IBD induction. The mucosal surface of the intestine plays a key role in the control of the host immune response by acting as a barrier against bacterial and dietary antigens, and it is strongly influenced by nutrients [34] (Figure 4).

#### 3.1.1. Dietary Risk Factors

The Western diet is enriched with sugars, fat (e.g., saturated, polyunsaturated n-6 fatty acids and cholesterol) and food additives (e.g., emulsifiers, colourants, preservatives, flavoring and processed carbohydrates). These latter compounds may act as xenobiotic into the gut and directly induce compositional and functional alterations of the gut microbioma, which impairs epithelial functions in the gut, that is, perturbs Paneth cells and the gut barrier homeostasis [35]. An excessive intake of refined sugars promotes dysbiosis and gut inflammation, but their role of in the development or progression of IBD has been poorly explored yet [29]. Emerging evidence indicates that the consumption of food additives perturbs microbial composition and promotes experimental gut inflammation; for example, some artificial sweeteners promote dysbiosis (with increased Bacteroides and reduced *Lactobacillus* spp.) [36]; likewise, emulsifiers perturb gut microbial community structure and promote susceptibility to gut inflammation. Maltodextrins are common dietary polysaccharides used as emulsifiers or energy source, and their role in the impairment of defense mechanisms and in the overgrowth of harmful bacteria (i.e., *E. coli* and *Salmonella* spp.) has been demonstrated in both in vitro and in vivo studies [37]. Furthermore, the administration of emulsifiers at a low concentration leads to low-grade inflammation and obesity/metabolic syndrome in wild-type mice, while it promotes colitis in IL-10^−/−^ mice [38]. Finally, food colourants Red40 (E129) and Yellow 6 (E110) metabolites (produced by commensals) promote colitis in mouse models [39].

A high red meat consumption is also a hallmark of the Western dietary pattern associated with an increased IBD risk [40]. The relative abundance of bile-tolerant anaerobes such as *Bacteroides*, *Alistipes* and *Bilophila* increased following consumption of animal-based protein, especially red and processed meat [41]. Dietary animal proteins enhance the sensitivity to experimentally induced gut inflammation, possibly by expansion of inflammatory genera such as *Escherichia, Streptococcus* and *Enterococcus*. Total fat and saturated fat suppressed richness and diversity of the gut microbiota, increasing anaerobic abundance [42]. For this reason, the high fat content of Western diet is believed to be a key factor in causing intestinal dysbiosis, [43] which promotes gut inflammation. It is noteworthy that obese patients have a more severe IBD course [44], which confirms the link between diet, obesity, activation of the innate immune system, inflammation, and organ dysfunction. Unsaturated fatty acids are essential components of the intestinal inflammatory response; indeed, they modulate cell membrane composition with a subsequent modification of the activation of receptors, for example, TLR-4 activation is facilitated by saturated fatty acids and inhibited by n-3 PUFAs (such as docosahexaenoic acid) [45]. Unsaturated fatty acids can also act as signaling molecules, targeting numerous membrane and nuclear receptors, and finally altering cellular gene expression in adipocytes, tissue macrophages and endothelial cells [46]. In conclusion, a Western dietary pattern may trigger a proinflammatory environment into the gut of susceptible individuals through alterations of the gut microbiome (leading to an upregulation of the TLR- and NOD-pathways) and the consequent impairment of the intestinal epithelial barrier functions. [18,47].

#### 3.1.2. Protective Dietary Factors

Unrefined complex carbohydrates rich in fibers (typically whole grains), and their bacterial metabolites exert a protective effect on the gut. Bacterial short chain fatty acids and especially butyrate maintain gut homeostasis by protecting intestinal barrier integrity and host immune responses. SCFAs indeed stabilize HIF-1 a transcription factor coordinating barrier protection [48], and probiotic supplementation of butyrate-producing bacteria improves epithelial barrier integrity in CD [49]. Moreover, butyrate exerts anti-inflammatory effects in the gut mucosa by inhibition of histone deacetylases and activation of G protein-coupled receptors present in gut epithelium and mucosal immune cells [50]. Glutamine and arginine are non-essential amino acids, but they are considered essential in stress situations, and both demonstrated anti-inflammatory properties in IBD models [51,52]. Glutamine is the preferential substrate for enterocytes, and its effect has been evaluated in numerous experimental models of gut barrier dysfunction. For instance, Zhang and colleagues [53] showed that glutamine supplementation was able to improve the growth performance of LPS-challenged broilers, alleviate the inflammatory response, reverse the deleterious effects of LPS on intestinal permeability and the integrity of intestinal mucosa barrier, downregulating the expression of molecules involved in TLR4/FAK/MyD88 signaling pathways [53]. Intraperitoneal injection of alanyl-glutamine suppressed Th17 cytokine production, reduced luminal chemokine secretion, and decreased macrophage infiltration to the peritoneal cavity in DSS colitis models [54]. Treatment with arginine or isoleucine also upregulated in vitro the secretion of human beta-defensin-1 [hBD1], an antimicrobial peptide capable of counteracting the microbial overgrowth of pathobiont species [55]. The amino acid tryptophan has also been involved in host–microbiome interactions. Tryptophan and its metabolites regulate intestinal homeostasis; for example, it has been observed that *Lactobacillus* spp. are able to switch their fuel from sugar to tryptophan to produce indoles [56]. The aryl hydrocarbon receptor is expressed on intestinal dendritic cells and lymphocytes and helps to maintain epithelial integrity via IL-22 secretion; this receptor has many ligands, some of which are present in foods. These include indole or tryptophan metabolites (derived from cruciferous vegetables), stilbenes (e.g., resveratrol in grape), carotenoids (present in yellow/orange/red vegetables) and flavonoids [56,57,58]. In fact, some phytochemicals are known to positively influence intestinal physiology, e.g., Curcumin (*Curcuma longa*) has shown anti-inflammatory properties in different experimental models of IBD [59] and in human clinical trials [60]. One of the proposed mechanisms for different phytochemicals is the inhibition of TLR-4 activation [61]. It has been demonstrated that a ginger compound, 1-dehydro- 10-gingerdione, acts on TLR-4 expression in macrophages, inhibits LPS binding to a TLR-4 co-receptor and downregulates TLR-4–mediated expression of NF-κB and the expression of inflammatory cytokines in LPS-treated macrophage cell lines [62].

Studying the effects of natural compounds on murine models of colitis and on mice subjected to high-fat diet (HFD), our team focused on the effects of geraniol, an acyclic monoterpene present in high percentages in citronella, rose and palmarosa essential oils. In mice with DSS-induced colitis, geraniol intake resulted in a strong reduction in weight loss, an improvement in stool consistency and stool blood content. Geraniol administered orally halved weight loss in mice and reduced the colitis activity index; geraniol administered by enema improved colitis symptoms by preserving the integrity of the colonic mucosa. These clinical observations were further supported by a significant reduction in mRNA expression of the pro-inflammatory enzyme COX-2 in the colic mucosa of geraniol-treated mice [63]. In HFD-fed mice, a food supplement based on D-Limonene (a monoterpene commonly found in the essential oils of *Citrus* plants), counteracted the metabolic negative effects of the diet by positively modulating the gut microbiota [64]. So, it can be stated with reasonable confidence that many different specific factors in food have direct effects on mucosal integrity and immune function, and these effects may overall be enclosed in the concept of nutrient signaling. In addition to these mechanisms, that may explain the effects of different nutrition patterns on IBD, diet is a major determinant for the intestinal microbiome and metabolome. It is not clear whether microbiome changes are the cause or an effect of IBD, but it is a matter of facts that the features of a Western diet pattern including higher caloric intake and consumption of sugar-sweetened beverages are negatively associated with microbiome diversity. In contrast, the features of a Mediterranean style diet with a higher consumption of fruits, vegetables and phytochemicals have been associated with an increased gut microbial diversity. The results of these studies have been relatively consistent, pointing to a lower risk of IBD among people who consume more fruits and vegetables, and a higher risk in people who consume more animal fats and refined sugars [65,66,67].

### 3.2. Malnutrition in IBD

There are two main definitions of malnutrition: the European Society for Clinical Nutrition and Metabolism defines it as a state resulting from a lack of nutrient intake or absorption, leading to altered body composition (decreased body fat mass and cellular mass) resulting in decreased physical and mental function and impaired clinical outcomes of disease [68]; the World Health Organization (WHO) defines malnutrition as a deficiency, excess or imbalance in a person’s energy/nutrient intake [69].

The prevalence of malnutrition among patients with IBD can vary from study to study, depending on the definition adopted and on the different screening tests used [70] and ranged from 20% to 85%. Malnutrition is more present in CD than in UC, probably because CD can affect any part of the gastrointestinal tract including the small bowel where most of the nutrient absorption occurs [71,72]. Malnutrition has been correlated with poor clinical outcome and quality of life in IBD patients and has also been documented in patients in clinical remission [73]. The main micronutrient deficiencies in patients with IBD are shown in Table 1. The mechanisms responsible for nutritional deficiencies are not always clear and may be related to the disease itself, which causes malabsorption, but also to increased metabolic demand (related to the active inflammatory process), and/or dietary habits of IBD patients, which influence their nutrient intake [74,75,76].

#### 3.2.1. Disease-Related Causes of Malnutrition

The nutritional status of patients with IBD can be altered due to the high catabolic load that can occur during the acute inflammatory phases of the disease. Inflammatory cytokines stimulate the release of glucagon, catecholamines and cortisol. This condition, without an adequate supply of nutrients, can lead to loss of weight, due to decreased subcutaneous adipose tissue and muscle mass [77].

Proinflammatory cytokines can also act as anorectic agents at the level of the intestine, where IL-1β is able to stimulate cholecystokinin release and potentiate its action [78], or in hypothalamic areas where TNF-α modulates glucose-sensitive neurons [79,80] affecting eating behavior [79,81,82].

These effects are consistent with the decreased appetite often experienced by IBD patients during the acute phase [83,84]. In addition to mechanisms directly stimulated by the inflammatory response, other common causes of malnutrition in IBD are gastrointestinal ulcerations and/or fistulas that lead to a functional or anatomical reduction in the gut absorbent surface [85]. Diarrhoea may cause a depletion of potassium, magnesium, phosphorus and zinc [86,87], while intestinal hemorrhages may cause iron deficiency [72,88]. Drugs used for the management of IBD such as sulfasalazine and methotrexate can cause a folic acid deficiency [89]. While in UC nutritional deficiencies are mainly related to diarrhoea and bleeding, in CD, surgeries, when the terminal ileum is extensively resected, can cause a significant reduction in the intestinal absorption surface area affecting nutrient assimilation including fats and fat-soluble vitamins (A, D, E, K) [90,91,92,93,94]. In CD patients, the reabsorption of bile salts could be impaired [95], and small intestinal bacterial overgrowth (SIBO) is a condition that often develops [96]. SIBO contributes to fat and vitamin B12 malabsorption due to bile acid deconjugation by luminal bacteria [97] and vitamin B12 consumption by bacterial metabolism [98].

#### 3.2.2. Food Habits-Related Causes of Malnutrition

The adoption of unbalanced diets by IBD patients is mainly attributable both to the Western diet pattern and to specific patient choices in the belief that certain food restrictions can prevent disease exacerbations or improve symptoms in the acute phase. Based on epidemiological studies, 48 to 51% of IBD patients believe that diet can initiate the disease [83,99,100], while 62 to 89% of patients self-impose dietary restrictions to try to avoid relapse or control symptoms [83,84,99,100]. Spices, fibers (fruit and vegetables), alcohol, carbonate beverages, milk and dairy products are the most commonly avoided foods by IBD patients, especially in the acute phase [83,84,99,100]. Furthermore, in the study conducted by de Vries and colleagues in 2019 [99], 30% ca. of IBD patients stated that they had never received any nutritional advice and that they preferred to eliminate certain elements rather than eat beneficial foods or follow a specific diet; moreover, among the patients who had received nutritional advice from professionals, 81% still based their restrictions on their own experience.

In a study conducted by Crooks and colleagues, one third of the patients interviewed reported having tried a specific exclusion diet such as a lactose-free, gluten-free, low FODMAPS (Fermentable Oligo-, Di-, and Monosaccharides and Polyols) diet, a specific carbohydrate exclusion diet, an anti-inflammatory diet and paleo diet [100]. The tendency of patients to self-manage their diet in the context of an extremely debilitating disease may have a negative impact on the development of eating disorders and certainly may worsen their nutritional status.

In a review published by Peters and collaborators in 2022 [101], authors found that the prevalence of eating disorders in patients suffering from gastrointestinal diseases ranged from 13 to 55%. Factors associated with these disorders included sex (female), younger age, gastrointestinal symptom severity, anxiety, depression and lower quality of life. In particular, the EAT−26 (eating attitude test-26) and BES (binge eating scale) scores were both higher in IBD vs. healthy subjects (20% and 25% above clinical threshold vs. 4% and 2%, respectively). Two studies have assessed the nutritional status of IBD patients undergoing food restriction: the first, published in 2013, was performed on 59 UC patients (56 of whom were in remission) and showed an inadequate intake of fiber, fat-soluble vitamins (vitamin A and E), vitamin C and minerals (calcium and magnesium), linked to the low-residue diet that is often followed during the acute phase, but also maintained in the remission phase [102]; the second study published in 2019 considered 104 IBD patients with both active disease and in remission, documenting severe malnutrition in 12.2% of patients on self-prescribed exclusion diets vs. 5.5% of patients on free diets [103]. Self-prescribed dietary restrictions were also associated with an inadequate calcium intake and low bone mineral density (in 80% and 51% of IBD patients, respectively) in a study performed on 90 patients [104].

Another study conducted on 54 CD patients with active disease and 30 healthy subjects reported that the former had lower intakes of fiber, vitamins (A, E, C, B6, folic acid) and β-carotene, as well as calcium, potassium, phosphorus, iron, magnesium, copper, and iodine. These patients also showed a lower dietary oxygen radical absorbance capacity (ORAC) linked to low consumption of fruits and vegetables, and a higher renal potential load (PRAL) linked to high consumption of red meat. In addition, blood concentrations of total cholesterol, potassium, iron and amino acids were significantly lower in CD patients compared to healthy subjects [105]. A similar study in pediatric CD patients in clinical remission showed lower fiber and vitamin A intake, along with higher animal protein consumption; also in this study, ORAC was lower, while PRAL was higher in CD patients compared to healthy subjects [106]. These data are consistent with epidemiological studies showing that parents of pediatric IBD patients pay particular attention to highly processed foods, fast foods, spicy foods and dairy products, but little attention to meat consumption, with a tendency to believe that fruits and vegetables may have a negative impact on the disease course [107,108].

### 3.3. Nutritional Interventions

In recent years, increasing attention has been paid to the study and validation of nutritional protocols aimed at inducing remission or symptom control in pediatric IBD patients in an attempt to avoid or delay the use of drug therapies such as corticosteroids or biologics. Specific carbohydrate diets (SCDs), Mediterranean (MD) or Mediterranean-like (MD-like) diets, exclusion diets (EDs) and low FODMAPS diets have been tested in clinical controlled trials on IBD patients, with promising results (Table 2).

The SCD was introduced way back in 1920 [109] for the management of celiac disease. It is based on the exclusion of refined sugars and the full restriction of complex carbohydrates including gluten-containing ones. To date, this diet is considered effective not only for the management of celiac disease, but also for IBD [110]. Small, controlled studies in pediatric IBD patients have shown encouraging results: two prospective studies evaluated the SCD in pediatric IBD patients with active disease, achieving clinical remission [111] and the decrease in both Harvey–Bradshaw Index (HBI) and Pediatric Crohn’s Disease Activity Index (PCDAI) [112].

A randomized controlled trial compared the efficacy of SCD, a modified version of SCD (MSCD, which includes consumption of oats and rice) and of the Whole Food diet (WFD, which excludes wheat, corn, sugar, milk and food additives) in achieving the clinical remission [113]. All three diets have been shown to induce clinical remission. Additionally, two retrospective studies, in which mainly CD patients with active disease were enrolled, showed clinical remission [114] or strong improvement in disease activity indices induced by the experimental diets (including MSCD and WFD) in almost all enrolled pediatric patients [115]. Different studies were performed on the effects of the Mediterranean diet (MD) in the management of IBD. A randomized trial on 194 IBD patients compared the SCD with the Mediterranean diet, showing very similar remission rates between the two diets [116]. Another study was conducted on the effects of MD on 184 adult patients (84 CD and 58 UC) with mild to moderate disease or in remission, monitored for six months; in addition to improvements in anthropometric and metabolic parameters, out of the 14 patients with active disease and stable therapy, 10 achieved remission and 4 achieved a significant decrease in disease severity, which shifted to a milder form [117]. In other studies, nutritional interventions were based on modified MD diets, characterized by the typical features of MD (MD-like diets); in particular, a randomized trial carried out on 7 CD patients with an active disease compared a high-fiber MD-like diet (HF, in which specific portions of whole grains were prescribed to patients) with an MD-like exclusion diet characterized by the elimination of whole grains, dairy products and spicy foods. The HF group showed better performances, with a decrease in Harvey–Bradshaw index score maintained after 4 weeks of follow-up [118]. In 17 patients with UC in remission, a MD-like low-fat, high-fiber diet (LFHF) was compared with an improved standard American diet characterized by a higher intake of vegetables, fruits, fish and dietary fiber. Results demonstrated that only the LFHF group showed a reduction in inflammatory markers after 4 weeks of the diet [119]. In a broader study, conducted on 214 subjects with CD in remission, patients were randomized to high consumption of red/processed meat (2 servings per week of red or processed meat) or a low meat MD-like diet (no more than 1 serving per month) and were monitored for 49 weeks; the disease relapse rate was 62% in the high meat consumption group versus 42% in the low meat consumption group [120]. Two types of exclusion diets have been tested on IBD patients: the IgG4-based exclusion diet and the Crohn’s disease exclusion diet (CDED). The IgG4 exclusion diet is based on the exclusion of four foods depending on IgG4 titration and is patient-specific; this diet was tested on CD patients with an active disease and controlled with a sham diet. After 4 weeks, there was an overall improvement in Crohn’s disease activity index in the group eating the IgG4-driven diet [121]. However, IgG4 positivity in blood seems to correlate with food exposure. Following an IgG4-driven diet primarily means changing eating habits, with a tendency towards a sharp reduction in the foods most frequently consumed.

CDED is a diet based on the elimination of specific foods considered “proinflammatory” and the introduction of mandatory or recommended “anti-inflammatory” foods. CDED was tested alone in 32 CD adult patients, showing an 82.1% efficacy in inducing clinical remission after 12 weeks [122], or combined with partial enteral nutrition (PEN) in a randomized trial comparing CDED vs. CDED + PEN, with results showing similar efficacy [123]. CDED+PEN showed a similar efficacy if compared to the exclusive enteral nutrition in a six-week study also on pediatric patients [124], but a superior efficacy was detected in a 12-week study carried out by Levine and colleagues [125]. Enteral nutrition is based on elemental (amino acid-based), semi-elemental (oligopeptides) and polymeric (whole protein-based) formulas that can be administered to IBD patients as a complete meal replacement (exclusive enteral nutrition) or in addition to standard meals (partial enteral nutrition). Its efficacy in inducing and maintaining remission has been supported by clinical trials and meta-analyses, despite the heterogeneity of treatment protocols [126]. One explanation for enteral nutrition efficacy is that the polymeric formulation could minimize antigen exposure in the gut resulting from food ingestion [127]; however, whether administered via nasogastric tube or ingested directly, enteral nutrition diets are often poorly tolerated by patients and can be used only for short periods of time. The combination of CDED and PEN, however, appears to be better accepted by patients [125].

The low oligo-, di-, monosaccharides and fermentable polyols (FODMAP) diet is characterized by the exclusion of short-chain fermentable carbohydrates, which are poorly digested by gut enzymes and thus predominantly fermented at the level of the intestinal microbiota. FODMAP content is high in fruits, vegetables, legumes, dairy products and wheat. FODMAP diet has been used with some success in patients with irritable bowel syndrome, a functional disorder characterized by abdominal pain, bloating and altered bowel habits [128]. Low FODMAP diet has been tested in several clinical trials in patients with IBD in remission or with a mild disease meeting the criteria for functional gastrointestinal disorders. In almost all studies, patients with CD or UC showed a general improvement in gastrointestinal symptoms [129,130,131,132,133], but there are insufficient data to reach conclusions about its efficacy in IBD management. Despite a rather large number of diets tested on rather limited numbers of IBD patients, these different diet regimens share some common features such as the exclusion of red and processed meats; reduction in saturated fats, sugars and processed foods; and an increase in the consumption of fiber, fruits and whole raw foods. Overall, these data provide clear guidance on the type of diet that should be recommended for patients with IBD, even when they are not inserted into specific dietary plans.

**Table 2 foods-12-01987-t002:** Interventional studies on IBD patients.

**Low FODMAPS Diet**						
**Authors**	**Study** **Design**	**Dietary** **Intervention**	**Patient** **Number**	**Clinical Activity**	**Days** **of Treatment**	**Results**
Cox et al., 2017 [129].	Randomized, double blind, placebo controlled, crossover.	LFD + 4 reintroduction challenge of fructans or galacto-oligosaccharides or sorbitol or placebo, each challenge preceded by 4 days of wash out.	12 CD 17 UC	Remission. All patients also met criteria for FGS.	3 days challenge.	Fructane reintroduction induced a worsening of FGS.
Bodini et al., 201. [130].	Randomized.	LFD vs. Free diet	35 CD 20 UC	Remission/mild disease. All patients also met criteria for FGS.	6 weeks	Decrease in HBI index and calprotectin but not in Mayo score in LFD group.
Pedersen et al., 2017 [132].	Randomized.	LFD vs. Free diet	61 UC 28 CD	Remission/mild disease. All patients also met criteria for FGS.	6 weeks	Decrease in IBS-SSS score in LFD, decrease in SCCAI in UC LFD Group, no change in HBI score.
Cox et al., 2020 [131].	Randomized.	LFD vs. Free diet	26 UC 26 CD	Remission. All patients also met criteria for FGS.	4 weeks	Decrease in IBS-SSS score in UC patients but not in CD patients.
Melgaard et al., 2022 [133].	Randomized, blinded placebo controlled.	LFD + reintroduction challenge	16 UC	Remission. All patients also met criteria for FGS.	8 weeks LFD + 2 weeks Low FODMAPs or Placebo	No effect on IBS-SSS score.
**Enteral Nutrition**						
**Authors**	**Study design**	**Dietary** **Intervention**	**Patient number**	**Clinical activity**	**Days** **of treatment**	**Results**
Guo et al., 2013 [134].	Pilot study.	EEN	13 CD	Active.	4 weeks	Clinical remission in 86% of patients.
Pigneur et al., 2019 [135].	Randomized.	EEN vs. steroids	19 CD, pediatrics	Active.	8 weeks	Same efficacy in inducing remission, 89% of mucosal healing in EEN arm, 17% of mucosal healing steroids arm.
Brückner et al., 2020 [136].	Open label	PEN vs. free Diet	41 CD, pediatrics.	Remission/mild.	12 months	Growth improvement.
Moriczi et al., 2020 [137].	Retrospective.	EEN	235 CD, pediatrics.	Active.	8 weeks	83% clinical remission.
**Specific Carbohydrate Diet (SCD)**						
**Authors**	**Study design**	**Dietary** **Intervention**	**Patient number**	**Clinical activity**	**Days** **of treatment**	**Results**
Lewis et al., 2021 [116].	Randomized.	SCD vs. MD	194 CD	Mild/ moderate.	6 weeks	Clinical Remission: 46.5% SCD, 43.5% MD.
Suskind et al., 2020 [113].	Randomized.	SCD vs. Modified SCD vs. WFD	18 CD, pediatrics.	Mild/ moderate.	12 weeks	Clinical remission in all arms.
Braly et al., 2017 [111].	Prospective, open label, non-controlled.	SCD	9 CD/UC, pediatrics.	Mild/ moderate.	12 weeks	Clinical remission.
Suskind et al., 2014 [114].	Retrospective.	SCD	7 CD, pediatrics.	Active.	3 months	Clinical remission.
Obih et al., 2016 [115].	Retrospective.	SCD	20 CD 6 UC pediatrics	Active.	6 months	Clinical remission in CD patients, decrease in PUCAI in UC patients.
Cohen et al., 2014 [112].	Prospective, non controlled.	SCD	10 CD, pediatrics	Active.	12 weeks	Decrease in HBI and PCDAI.
**Mediterranean Diet/Mediterranean-like**						
**Authors**	**Study** **Design**	**Dietary** **Intervention**	**Patient number**	**Clinical** **activity**	**Days** **of treatment**	**Results**
Chicco et al., 2021 [117].	Prospective, non controlled.	MD	84 UC, 58 CD	Mild/moderate and remission.	6 months	Remission in patients with active disease.
Brotherton et al., 2014 [118].	Randomized, controlled, single blind.	HF vs. Exclusion Diet	7 CD	Active.	4 weeks	Decrease in HBI score in HF group.
Albenberg et al., 2019 [120].	Randomized.	HM vs. LM	214 CD	Remission.	49 weeks	Relapse in 62% HM group, 42% LM group.
Fritsch et al., 2021 [119].	Randomized, crossover.	LFHF vs. iSAD	17 UC	Remission/mild.	4 weeks	LFHF decreased markers of inflammation.
**Exclusion Diets**						
**Authors**	**Study** **Design**	**Dietary** **Intervention**	**Patient number**	**Clinical activity**	**Days** **of treatment**	**Results**
Guasekeera et al., 2016 [121].	Randomized, controlled.	IgG4-guided diet vs. sham diet	98 CD	Active.	4 weeks	Improvement of CDAI in the IgG4-guided diet.
Szczubełek et al., 2021 [122].	Prospective, non-controlled.	CDED	32 CD	Active.	12 weeks	Clinical remission in 82.1% of cases.
Yanai et al., 2022 [123].	Randomized, open label.	CDED + PEN vs. CDED	44 CD	Mild/ moderate.	24 weeks	Clinical remission in 68% of patients undergoing CDED+ PEN, and in 57% of patients undergoing CDED.
Levine et al., 2019 [125].	Randomized.	PEN+CDED vs. EEN followed by PEN	78 CD, pediatric.	Mild/ moderate.	12 weeks	Remission: 75,6% in CDED +PEN arm, 45.1% EEN+ PEN arm.
Sigall et al., 2021 [124].	Randomized.	EEN vs. PEN + CDED	73 CD, pediatric.	Mild/ Moderate.	6 weeks	Remission: 61.5% in CDED+PEN arm, 64.7% in EEN arm (at week 3).

LFD: Low FODMAPs Diet; EEN: Exclusive Enteral Nutrition; PEN: Partial Enteral Nutrition; SCD: Specific carbohydrate diet; MD: Mediterranean diet; WFD: Whole Food Diet; HF: High Fiber, HM: High in take in red Meat; LFHF: Low Fat High Fiber, iSAD: improved Standard American, CDED: Crohn’s Disease Exclusion Diet. FGS: Functional Gastrointestinal Disorders. HBI: Harvey-Bradshaw Index; IBS-SSS: Irritable Bowel Syndrome -Severity Scoring System; SCCAI: Simple Clinical Colitis Activity Index; PUCDAI: Pediatric Ulcerative Colitis Activity Index; PCDAI: Pediatric Crohn’s Disease Activity Index; CDAI: Crohn’s Disease Activity Index.

The main features of nutritional interventions that have been implemented in IBD patients are shown in Table 3.

## 4. Conclusions

The rapidly increasing incidence of IBD and other immune-mediated diseases points to a role of environmental factors in their pathogenesis; in particular, there is growing evidence that diet plays a key role in the development of IBD. The complexity of the human diet makes it very difficult to identify single foods as risk factors; indeed, it is more likely that complex nutritional patterns may have an impact on the pathogenesis and clinical course of IBD. In evolutionary terms, people in industrialized countries can be described as nutritionally maladapted and thus more susceptible to many diet-related and gut-related noncommunicable diseases, including IBD.

Once the disease has developed, IBD patients with active disease tend to adopt unbalanced diets, probably as an indirect response to their symptoms, that contribute to a reduced intake of certain nutrients with antioxidant, anti-dysbiotic and anti-inflammatory effects. Nevertheless, most of their nutrient deficiencies depend on impaired intestinal absorption caused by the disease itself. However, the role that diet can play in supporting conventional therapies for IBD is too often underestimated since it may help to induce and maintain clinical remission, but also improve patients’ quality of life. As a matter of fact, there are clear indications that diet can significantly modulate disease onset and activity, but unfortunately, there are no specific dietary guidelines for patients with IBD. This leaves the choice to the patients for choosing do-it-yourself diets which tend, in general, to reduce protective foods such as those of vegetable origin in favor of meat, especially red ones. IBD patients should be referred to a nutritionist for a counseling program soon after diagnosis to avoid do-it-yourself attempts to improve their nutrition. The diet model to be preferred for IBD patients should be a Mediterranean or Mediterranean-like diet including the traditional Asian ones, with a strong presence of raw foods of vegetable origin, fish and a reduced intake of meats (especially red and processed) and industrialized processed foods. In patients in whom the response to this type of diet is not satisfactory, an approach based on CDED + PEN or CDED alone can be adopted, certainly more complex to implement, but which seems to have greater efficacy in inducing remission or in maintaining it.

IBD patients, being at significant risk for malnutrition, should be screened for nutrient deficiencies using validated tools, and should receive nutritional advice and undergo oral, enteral or parenteral nutritional supplementation. However, the management of malnutrition in IBD is complex, and future clinical studies are needed to provide guidance to standardize the diagnosis and management of malnutrition in these patients.

## Figures and Tables

**Figure 1 foods-12-01987-f001:**
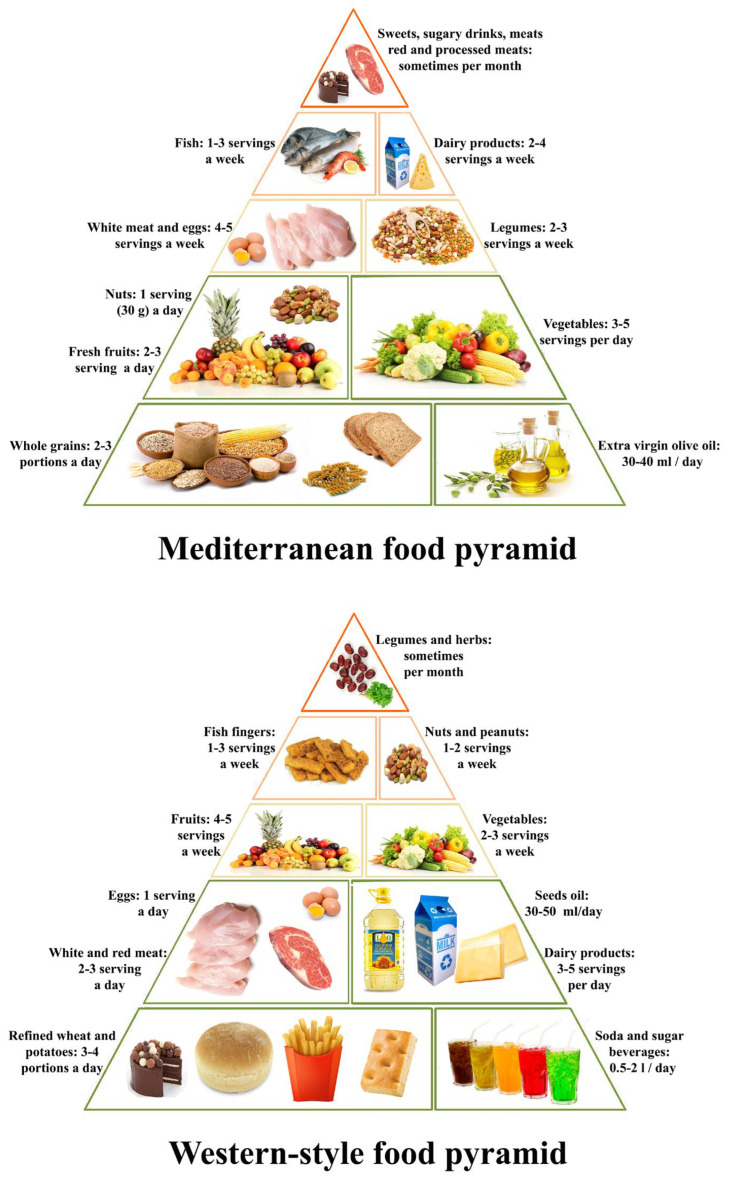
Comparison between the Mediterranean diet (**upper**) and the Western-style diet (**lower**) pyramids [17].

**Figure 2 foods-12-01987-f002:**
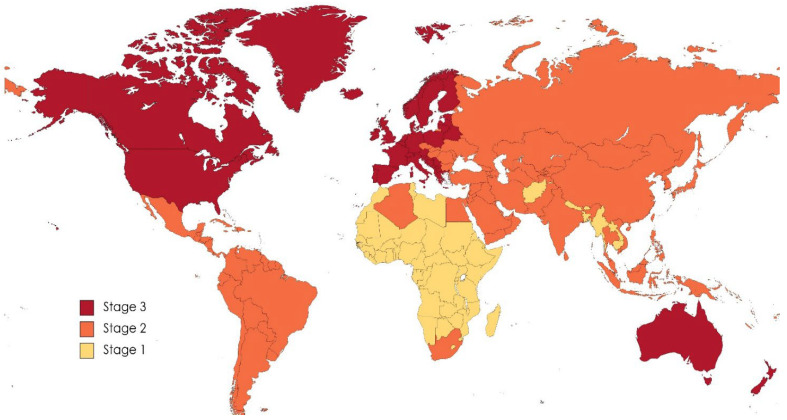
Epidemiological stages of IBD in the world. Stage 1: emergence. Stage 2: acceleration of incidence. Stage 3: compounding prevalence [4]. Developed economies: United States, Canada, Japan, Australia and New Zealand, Europe (except Southeastern Europe). Economies in transition: The Commonwealth of Independent States and Georgia, Southeastern Europe. Developing economies: Africa, Asia (except Japan), Latin America and the Caribbean (modified from [4]).

**Figure 3 foods-12-01987-f003:**
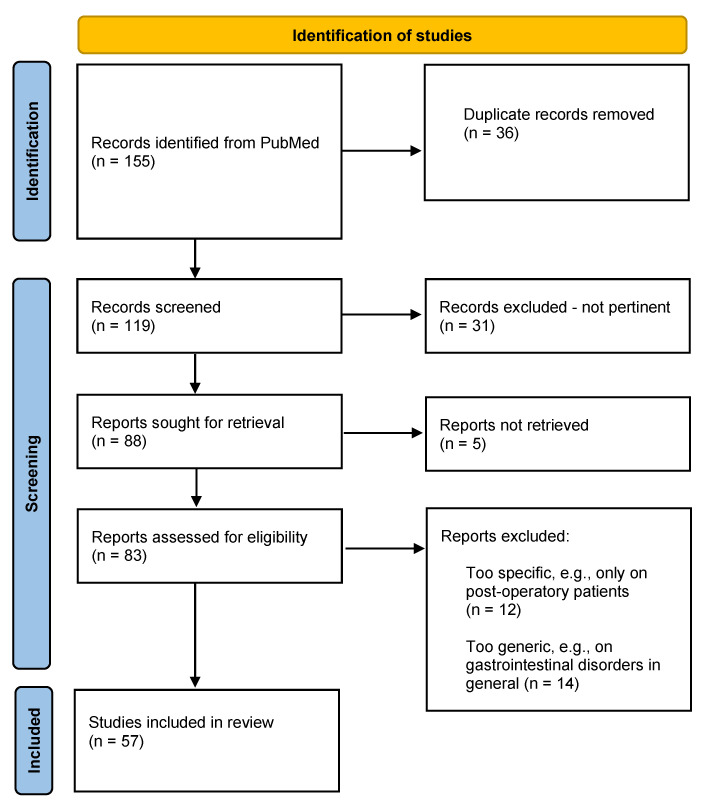
PRISMA Flow Diagram.

**Figure 4 foods-12-01987-f004:**
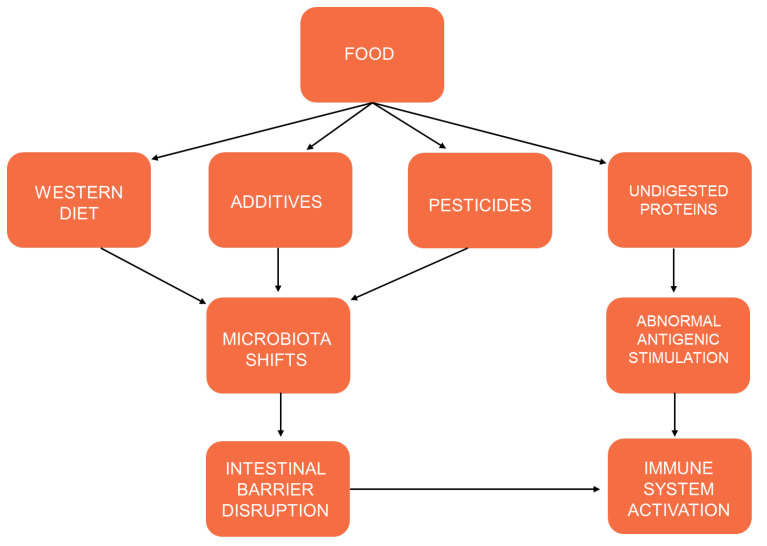
Interactions between food, environmental factors, intestinal microbes and immune system in IBD pathogenesis.

**Table 1 foods-12-01987-t001:** Prevalence (%), causes and effects of micronutrient deficiency in IBD patients.

Micronutrient Deficiency Prevalence (%)				
	UC	CD	Causes of Deficiencies *	Main Effects
Folic acid	35	54–67	GI resections; use of sulfasalazine, methotrexate, cholestyramine; achlorhydria; small intestinal bacterial overgrowth (SIBO).	Cause and worsen anemia, hyperhomocysteinemia (causes thromboembolic events), risk of carcinogenesis.
Vitamin B12	5	48	Terminal ileitis, resection if the terminal ileum > 20 cm, small intestinal bacterial overgrowth (SIBO).	Exacerbate anemia, hyperhomocysteinemia (causes thromboembolic events).
Vitamin C	16	24	Malabsorption, polymorphisms in genes encoding vitamin C transporters. TNF-α also downregulates transcription of transporters necessary for vitamin C uptake.	Impaired uptake and utilization of iron, wound healing deficit, bleeding.
Vitamin A **	26–93	11–50	Terminal ileum disease or ileal resection (malabsorption).	Associated with severity of the disease, wound healing deficit.
Vitamin D **	40	70	Terminal ileum disease or ileal resection (malabsorption), small intestinal bacterial overgrowth (SIBO)	Calcium deficiency, bone mineral density loss, compromised mucosal barrier functions, associated with disease activity, risk of carcinogenesis.
Vitamin E **	5	5	Terminal ileum disease or ileal resection (malabsorption).	Lipid peroxidation, reduced wound healing.
Vitamin K **	44	54	Terminal ileum disease or ileal resection (malabsorption).	Bone mineral density loss, bleeding.
Iron	81	39	Disease in/resection of the proximal small bowel (limited intestinal absorption), rectal bleeding, achlorhydria, small intestinal bacterial overgrowth (SIBO).	Anemia, fatigue, abnormal growth and inadequate cognitive development in children and adolescents.
Potassium	NA	6–20	Colectomy/pouch, prednisone, diarrhoea, vomiting, mucosal inflammation (increased secretion).	Muscle weakness/cramps, cardiac arrhythmia.
Calcium	10	13	Disease in/resection of the proximal small bowel (limited intestinal absorption), corticosteroid therapy, disturbed metabolism of vitamin D.	Bone mineral density loss, osteoporosis, risk of carcinogenesis.
Magnesium	NA	14–88	Chronic or severe acute diarrhoea, short gut (reduced intestinal absorption), disturbed metabolism of vitamin D.	Bone mineral density loss.
Zinc	38-45	40-50	Diarrhoea, ostomies, high-exit fistulas, chronic malabsorption state due to intestinal inflammation.	Oxidative cellular damage, wound healing deficit, dysfunctional epithelial barrier, altered mucosal immunity, increased pro-inflammatory cytokines.
Selenium	NA	35–40	Impaired intestinal absorption (duodenum, caecum).	Inflammation, risk of carcinogenesis.

* Besides the reduced intake, ** fat soluble vitamins. NA: not available.

**Table 3 foods-12-01987-t003:** Types of nutritional interventions in IBD patients.

**Low Fermentable Oligosaccharides,** **Disaccharides, Monosaccharides, and Polyols (FODMAP) Diet**	
**Allowed foods**	Eggs and meat. Certain cheeses such as brie, camembert, cheddar and feta. Almond milk, rice, quinoa and oats. Vegetables such as eggplant, potatoes, tomatoes, cucumbers and zucchini. Fruits such as grapes, oranges, strawberries, blueberries and pineapple.
**Restricted foods**	Dairy-based products, wheat-based, beans and lentils. Vegetables, such as artichokes, asparagus, onions and garlic. Fruits such as apples, cherries, pears and peaches.
**Other Details**	This diet consists of three steps: (1) elimination of all FODMAPs for 4–6 weeks, (2) gradual reintroduction aimed to individuate foods related to symptomatology, (3) personalization of diet.
**Enteral Nutrition**	
**Allowed foods**	Elemental formulas (amino acids), semi-elemental formulas (oligopeptides), or polymeric formulas (whole proteins).
**Restricted foods**	All other foods.
**Other Details**	Enteral nutrition can be administered both by nasogastric tube or ingested. Enteral nutrition can be “exclusive” when formulas completely substitute meals, or “partial” when it is administered together with meals. Partial enteral mutrition is often prescribed in association to Crohn’s disease exclusion diet.
**Specific Carbohydrate Diet (SCD)**	
**Allowed foods**	Additive-free meats, poultry, fish and shellfish. Additive-free and sugar-free oils, white vinegar, cider and mustard. Additive-free and sugar-free coffee, tea and fruit juice. All-natural, sugar-free peanut butter. Cheeses such as sharp cheddar, colby, swiss and dry curd cottage cheese. Fresh, frozen, raw or cooked vegetables, including string beans. Fresh fruits or frozen, cooked or dried fruits without added sugar. Eggs. Homemade yogurt that ferments for at least 24 h. Honey. Legumes such as dried navy beans, lentils, peas, split peas and lima beans. Also, unroasted cashews and unroasted peanuts in the shell. Nuts, peanuts and nut flours.
**Restricted foods**	Grains such as barley, corn, oats, quinoa, rice and wheat. Grain products such as bread, cereal and pasta. Candy, chocolates and other products made with sugar, high fructose corn syrup, or fructo-oligosaccharides (FOS). Canned or processed meats. Canned vegetables with additives. Certain legumes such as soybeans, chickpeas and bean sprouts. Dairy products high in lactose such as mild cheddar, store-bought yogurt, milk, cream, ice cream and sour cream. Powdered spices such as curry, garlic and onion. Seaweed. Starches such as potatoes, sweet potatoes and turnips. Sugars including molasses, corn syrup, maple syrup, fructose, sucrose and other processed sugars.
**Mediterranean Diet**	
**Allowed foods**	Vegetables, fruits, cereals, nuts, legumes, unsaturated fat such as olive oil. Medium intake of fish, dairy products, wine.
**Restricted foods**	Saturated fat, meat, processed foods, processed meat and sweets
**Crohn’s Disease Exclusion Diets**	
**Allowed foods**	Phase I Mandatory daily foods: fresh chicken breast 150–200 g, 2 eggs, 2 bananas, 1 fresh apple, 2 potatoes (potatoes must be cooked and refrigerated before use). Allowed daily foods: fresh strawberries, fresh melon (1 slice), rice flour, white rice and rice noodles (unlimited), 2 tomatoes (additional allowed for cooking), 2 cucumbers (2 medium size), 2 avocado halves, 1 arrot, spinach 1 cup uncooked leaves, lettuce (3 leaves), onion, fresh green herbs (e.g., basil, parsley, coriander, rosemary, thyme, mint, dill), 1 glass freshly squeezed orange juice from fresh oranges (not from cartons or bottles), water, sparkling water, salt, pepper, paprika, cinnamon, cumin, turmeric, 3 tablespoons of honey, 4 teaspoons of sugar, fresh ginger and garlic cloves, lemons and limes. Foods allowed only once a week: fresh lean fish (not deep fried, dietitian guidance required). Phase II Mandatory daily foods: fresh chicken breast 150–200 g, 2 eggs, 2 bananas, 1 fresh apple, 2 potatoes (potatoes must be cooked and refrigerated before use). Allowed daily foods: fresh strawberries, fresh melon (1 slice), Rice flour, White rice and rice noodles (unlimited), 2 Tomatoes (additional allowed for cooking), 2 Cucumbers (2 medium size), 2 Avocado halves, 1 Carrot, 1 cup uncooked spinach leaves, lettuce (3 leaves), onion, fresh green herbs (e.g., basil, parsley, coriander, rosemary, thyme, mint, dill), 1 glass freshly squeezed orange juice from fresh oranges (not from cartons or bottles), water, sparkling water, salt, pepper, paprika, cinnamon, cumin, turmeric. 3 tablespoons of honey, 4 teaspoons of sugar, fresh ginger and garlic cloves, one slice whole grain bread daily, quinoa, 3 tablespoons of cooked lentils or peas, 6 almonds or walnut halves (unprocessed), baking soda. Foods allowed only once a week: fresh lean fish (not deep fried, dietitian guidance required), 200 gr Sirloin or fillet steak (maximum), 1 slice whole grain bread (maximum), 1 can of tuna (in olive or canola oil) drained, ½ cup of oatmeal or cut oats. Additional daily foods from week 7: broccoli, cauliflower 2 florets daily, 4 fresh mushrooms (not canned), ½ red bell pepper, 1 zucchini or slice squash, 1 pear or kiwi or ripe nectarine. Additional daily foods from week 10: most vegetables (restricted amounts with dietitian guidance), most fruits (restricted amounts with dietitian guidance), quinoa, 3–4 tablespoons of cooked lentils or peas.
**Restricted foods**	Dairy, animal fat, wheat, emulsifiers, artificial sweeteners, other cuts or parts of chicken, other sources animal or soy protein, carrageenan, maltodextrins (and sucralose), sulfite containing foods, xanthan gum, packaged/canned/frozen precooked foods, doughs, baked goods, frozen, canned fruits and vegetables, oral iron supplements, soy or gluten-free products, ready to use sauces, syrups, spreads dressings, margarine, butter, vinegar, soy sauce, ketchup, mayonnaise, alcoholic beverages, soft drinks, juices, deep-fried or oily foods.
**Other Details**	Each phase has a duration of 6 weeks. CDED is often associated with polymeric formulas which, during phase 1, represent 50% of the caloric intake. Fruit and vegetables are progressively reintroduced during phase II.
**IgG 4-guided diet**	
**Allowed foods**	Foods not associated with IgG4 reactivity.
**Restricted foods**	Foods associated with IgG4 reactivity.
**Other Details**	IgG4-guided exclusion diet is a personalized approach excluding and replacing in each patient 4 food types with the highest IgG4 titration. Screening is performed on 16 food types: milk, peanuts, soya, shrimp, egg, tomato, pork, beef, cod fish, potato, wheat, yeast, cheddar cheese, chicken, lamb and rice.

## Data Availability

Data available in a publicly accessible repository. The data presented in this study are openly available in the Pub Med database.

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
