# Peer review of "The Bidirectional Link between Nutritional Factors and Inflammatory Bowel Diseases: Dietary Deficits, Habits, and Recommended Interventions—A Narrative Review"

_foods, 2023, doi:10.3390/foods12101987_

Round 1

Reviewer 1 Report

The present paper is an interestingly approaching to an actual nutrition issue; it investigates the nutritional approach towards inflammatory bowel diseases under the form of a narrative (non-systematic) review screening
PubMed literature on keywords (((inflammatory bowel disease) AND (diet*ther* OR nutri*inter*)) OR ((inflammatory disease AND (nutri*deficienc* OR malnutrition))); (((inflammatory bowel disease) AND ((diet) OR (western diet) OR (nutri*))). In the form of a clinical trial, basic research in the English language, with full text available (?), published during the last ten years (2012-2022?).

The search resulted in in155 articles, from which 83 abstracts were selected, and 57 most relevant studies were selected for the complete review, with the main conclusion that in the context of no validated tool for screening for malnutrition in IBD patients and no standardized nutritional regimen for these patients "future studies are needed to provide guidance to standardize the diagnosis and management of malnutrition in IBD.

The authors have done an excellent job reviewing most of the existing literature on a general topic of massive interest for clinical practice, research, and academia. The paper approaches (1) Food-related pathogenetic molecular mechanisms of IBD - Dietary risk factors and Protective dietary factors (2) Malnutrition in IBD - Disease-related causes of malnutrition, and Food habits-related causes of malnutrition. (3) Nutritional interventions Low FODMAPS Diet, Enteral nutrition, Specific carbohydrate diet, Mediterranean diet/like, Exclusion diets, with a centralization of the diets found in the studied literature in the form of a food permission table, ready to be used in practice for every type of the above-listed diets.

However, some issues can be improved.

1.     Title

"Diet in IBD: the gap between how it should be and how it usually is"

·       Using an acronym for intestinal bowel disease restrict the readers to specialists,

·       The expression of the title is too general – a more specific or concrete title would be welcomed (e.g., Review of the diet in inflammatory bowel disease "or "The bidirectional link between nutritional factors and IBD. Dietary deficits, habits, and recommended interventions"), at author's latitude.

2.     Keywords: Inflammatory Bowel Diseases, Crohn's Disease, Ulcerative Colitis, Diet, malnutrition, nutritional interventions

"enteral nutrition" may be an option to be included as a keyword.

3.     Abstract

·       Line 34: (Abstract) conclusions should cover the research question (present also in the title) – "Diet in IBD how it should be and how it usually is"

  • Please review the English language correctness and character number
  • The methodology of the study may be more detailed (keywords and number of articles examined)

4.     References

  • Most of the references are actual. However, several references are older than ten years.
  • Please check the correctness of the format

1.     Introduction

·       Line 124 – are the images proprietary or referenced? If so, please provide the source

·       The introduction reveals the actual knowledge on the topic (diets, the good and bad, FAo survey on the nutrition transition) courteously; however, the unknown of the research is less marked in this section, being actually "no validated tool for screening for malnutrition in IBD patients and no standardized nutritional regimen for these patients" mentioned in the abstract.

The purpose of the research may derive from it; any other purpose of the study should be optimally inserted here. A specific knowledge gap would enhance the attention of the audience.

5.     Materials and methods

·       even if not a Systematic review 'The quality of a narrative review may be improved by borrowing from the systematic review methodologies that are aimed at reducing bias in the selection of articles for review and employing an effective bibliographic research strategy (https://www.tandfonline.com/doi/abs/10.1179/2047480615Z.000000000329')

·       Pubmed is a search engine (interface) used to search Medline and other databases. Could the authors comment on what specific databases were accessed (Medline, PubMed Central, etc.)?

·       Please mention the city and country of the database search resource used.

6.     Results

·       Line 166 - Study selection subtopic

1.     even if not a Systematic review, a graphical illustration of the study search would make the research easier to follow. Detailed information regarding the selection process might be presented in a PRISMA-type flow diagram

2.     in a fragment referring to the search results, the authors might mention the list of the analyzed topics as retrieved by the search and as presented in the paper as (1) Food-related pathogenetic molecular mechanisms – dietary risk factors, dietary protective factors; (2) malnutrition in IBD – macro and micronutrient deficiency prevalence in IBD; disease-related causes of malnutrition in IBD, food habits related causes of malnutrition in IBD; (3) nutritional interventions in IBD

7.     Conclusions

  • the main results of the study, followed by authors' recommendations derived from the study result to be mentioned here in the results section and subsections in order
  1. The abbreviation section, even if optional, would improve the clarity of the reading.

Author Response

Dear Collegue,

Thank you for your revision.

We have followed all your helpful suggestions.

Please see the following review followed point by point.

  1. Title

"Diet in IBD: the gap between how it should be and how it usually is"

  • Using an acronym for intestinal bowel disease restrict the readers to specialists,
  • The expression of the title is too general – a more specific or concrete title would be welcomed (e.g., Review of the diet in inflammatory bowel disease "or "The bidirectional link between nutritional factors and IBD. Dietary deficits, habits, and recommended interventions"), at author's latitude.

R: We thank the reviewer for these comments: title has been changed as suggested

  1. Keywords: Inflammatory Bowel Diseases, Crohn's Disease, Ulcerative Colitis, Diet, malnutrition, nutritional interventions

"enteral nutrition" may be an option to be included as a keyword.

R: keyword was added.

  1. Abstract
  • Line 34: (Abstract) conclusions should cover the research question (present also in the title) – "Diet in IBD how it should be and how it usually is"

R: Abstract was amended as recommended

  • Please review the English language correctness and character number

R: done

  • The methodology of the study may be more detailed (keywords and number of articles examined)

R: methodology was amended as suggested

  1. References
  • Most of the references are actual. However, several references are older than ten years.

R: only articles published in the last 10 years were analyzed in this review. Bibliographical notes going back more than 10 years are references reported in other cited articles, or were used for the introduction and discussion sections.

  • Please check the correctness of the format

R: Done. References are written as required by MDPI

  1. Introduction
  • Line 124 – are the images proprietary or referenced? If so, please provide the source

R: This image was used also in a previous paper published by our team. Reference was added.

  • The introduction reveals the actual knowledge on the topic (diets, the good and bad, FAo survey on the nutrition transition) courteously; however, the unknown of the research is less marked in this section, being actually "no validated tool for screening for malnutrition in IBD patients and no standardized nutritional regimen for these patients" mentioned in the abstract.

The purpose of the research may derive from it; any other purpose of the study should be optimally inserted here. A specific knowledge gap would enhance the attention of the audience.

R: We thank the reviewer for the advice. Introduction was amended accordingly.

  1. Materials and methods
  • even if not a Systematic review 'The quality of a narrative review may be improved by borrowing from the systematic review methodologies that are aimed at reducing bias in the selection of articles for review and employing an effective bibliographic research strategy (https://www.tandfonline.com/doi/abs/10.1179/2047480615Z.000000000329')
  • Pubmed is a search engine (interface) used to search Medline and other databases. Could the authors comment on what specific databases were accessed (Medline, PubMed Central, etc.)?

R: done

  • Please mention the city and country of the database search resource used.

R: done

  1. Results
  • Line 166 - Study selection subtopic
  1. even if not a Systematic review, a graphical illustration of the study search would make the research easier to follow. Detailed information regarding the selection process might be presented in a PRISMA-type flow diagram

R: we thank the reviewer for the advice. A PRISMA diagram was inserted in “Materials and methods” section.

  1. in a fragment referring to the search results, the authors might mention the list of the analyzed topics as retrieved by the search and as presented in the paper as (1) Food-related pathogenetic molecular mechanisms – dietary risk factors, dietary protective factors; (2) malnutrition in IBD – macro and micronutrient deficiency prevalence in IBD; disease-related causes of malnutrition in IBD, food habits related causes of malnutrition in IBD; (3) nutritional interventions in IBD

R: We thank the reviewer for the advice. The paragraph was amended accordingly.

  1. Conclusions
  • the main results of the study, followed by authors' recommendations derived from the study result to be mentioned here in the results section and subsections in order

R: as suggested, a paragraph summarizing the results of the study and our recommendations have been added to the conclusion section.

  1. The abbreviation section, even if optional, would improve the clarity of the reading.

            R: done

Reviewer 2 Report

I read with great interest the review article by Saracino and her colleagues.

I reviewed the manuscript mainly in terms of its methodology and less from a clinical point of view. Therefore, my comments focus on the methodological area.

The methodology section is very short. The researchers did not clearly specify which rules they used to disqualify articles (they stated that they located 155 articles and of those they reviewed only 57 in the end). Without detailing the rules for the selection of the articles, it is possible that there is a systematic bias in the selection. It is also suggested to add a clear flow chart of the selection of the articles.

It was expected that in order to answer the research topic, observational articles (such as cohort studies and case-control studies) would also be reviewed. The authors completely ignored this type of research. Why?

It is expected that a systematic literature review will be based on the accepted method of reporting as described in the PRISMA checklist.

In addition, it is expected that the GRADE method (Grading of Recommendations, Assessment, Development, and Evaluation) will be included in the methods section and accordingly in the critical review of the collected articles.

In my opinion, after making the above mentioned changes from a methodological point of view, the article will be considered.

Author Response

Dear Collegue,

Thank you for your revision.

We have followed all your helpful suggestions.

Please see the following review followed point by point.

The methodology section is very short. The researchers did not clearly specify which rules they used to disqualify articles (they stated that they located 155 articles and of those they reviewed only 57 in the end). Without detailing the rules for the selection of the articles, it is possible that there is a systematic bias in the selection. It is also suggested to add a clear flow chart of the selection of the articles.

It was expected that in order to answer the research topic, observational articles (such as cohort studies and case-control studies) would also be reviewed. The authors completely ignored this type of research. Why?

R: In the section “Food habits-related causes of malnutrition” we actually considered also surveys and observational studies. Materials and methods paragraph was amended.

It is expected that a systematic literature review will be based on the accepted method of reporting as described in the PRISMA checklist.

R: We thank the reviewer for the advice. This is a narrative review; however, we agree that a Prisma diagram would make the research easier to follow. The diagram was inserted in “materials and methods" section.

In addition, it is expected that the GRADE method (Grading of Recommendations, Assessment, Development, and Evaluation) will be included in the methods section and accordingly in the critical review of the collected articles.

R: We thank the reviewer for this advice. Since this is a narrative review, we did not structured the research and data analysis in such a way, so it is not possible for us to provide this type of information/assessment.

Round 2

Reviewer 2 Report

The authors answered most of the comments except for one comment that is extremely significant.

The researchers stated that they did not grade the manuscripts on which their research was based using accepted methods (such as GRADE). This point weakens the findings of the study and the researchers must state this as a limitation of the study.

Another thing is that the researchers stated in their answer that this is a narrative review. This type of research appears only once in the manuscript on page 5. This is a very important point, so it should be noted in the title of the manuscript.

Author Response

Dear Collegue,

Thank you for your revision.

We have followed all your helpful suggestions.

Please see the following review followed point by point.

The authors answered most of the comments except for one comment that is extremely significant.

The researchers stated that they did not grade the manuscripts on which their research was based using accepted methods (such as GRADE). This point weakens the findings of the study and the researchers must state this as a limitation of the study.

R: we thank the reviewer for this comment: we stated this point in materials and methods section

Another thing is that the researchers stated in their answer that this is a narrative review. This type of research appears only once in the manuscript on page 5. This is a very important point, so it should be noted in the title of the manuscript.

R: we thank the reviewer for this comment: title has been changed as suggested